# Attention to Bayes: Self-Attention and Bayesian Methods for Predicting 3D Chromatin Folding From DNA Sequences

## Abstract

We present a novel model and approach, combining Transformers and VAE-like model that builds upon the established Akita/Basenji CNN architecture to enable exciting new possibilities of predicting 3D chromatin folding from DNA sequences. We establish the motivation for using self-attention and a simple yet feasible Bayesian extension, derived from the usual posterior predictive distribution. Finally, we analyse the model, demonstrate the benefits and novelty of our approach, and discuss meaningful next steps.

## 1 Introduction

Gene expression is the process by which instructions encoded in DNA are converted into functional products such as proteins and RNA. The three-dimensional folding of the genome is essential for precise gene regulation. In mammalian interphase, chromatin structure is characterized by higher-order organizational units like chromatin loops, topologically associating domains (TADs), A/B compartments, and chromosome territories (Kumar et al. [2021]), which collectively compartmentalize the genome and bring distant functional elements into closer spatial proximity (Brown [2003]). Recent research suggests that loop extrusion involving cohesin and the orientation-specific binding of CTCF, an 11 zinc finger protein, is a key mechanism for the formation of these architectural units (Banigan and Mirny [2020]). Crucially, the deformation of TAD boundaries and enhancer-promoter loops has been implicated in gene dysregulation, cancer (Krijger and de Laat [2016]), and neurologic disorders such as schizophrenia and Alzheimer's Disease (Behrends and Engmann [2021]). Therefore, it is important to understand the determinants of 3D chromatin configuration and the impact of structural perturbations on gene expression.

Up to now, wet-lab chromatin conformation capture techniques such as 3C, Hi-C, and Micro-C have been developed to describe the chromatin architecture through genome-wide chromosome conformation capture (Lieberman-Aiden et al. [2009]). These methods use proximity ligation to produce a symmetrical contact matrix representation of pairs of chromatin interactions and enable simultaneous experimental mapping of locus-specific contacts across the whole genome. However, the construction of high-resolution Hi-C maps that allow for the detection of ultra-fine loop structures could theoretically require billions of ligation products, so Hi-C remains relatively expensive (Golov et al. [2020]).

There are several recent models that focuses on chromatin structure prediction from DNA sequences, namely DeepTACT (Li et al. [2019]), SEPT (Jing et al. [2020]), DeepC (Schwessinger et al. [2020]) and Akita (Fudenberg et al. [2020]). These models mainly utilize convolutional neural networks for the prediction task. However, chromatin interaction involves long-distance dependency which CNN models may not be able to effectively capture. This brought us the thought that we might be able to benefit from the attention mechanism since it is able to model for dependencies between any pairs of elements in a given sequence. (Vaswani et al. [2017])

## 1.1 Akita

One of the most successful models at predicting three-dimensional genome folding from DNA sequences is the Akita model (Fudenberg et al. [2020]). Akita is a convolutional neural network (CNN) model that uses only DNA sequences as inputs and transforms them into two-dimensional frequency maps of chromatin contact sites. In the model, the Basenji architecture (Kelley et al. [2018]) acts as the "trunk" of Akita, taking in 1Mb of 1-hot encoded DNA, applying one-dimensional convolutional blocks and dilated residual convolutions, and forming two-dimensional maps from these inputs in order to predict contact maps for five high-quality Hi-C and Micro-C datasets.

The model is able to make predictions related to CTCF binding and genome accessibility, reveal large effects of CTCF motif disruptions on genome folding, enable cross-species analyses of genome folding and learn species-specific relationships between DNA sequence and genome folding. Akita's performance is quantified with Mean Squared Error (MSE), Pearson's R, and Spearman's R of observed and expected maps on 1-Mb test set regions. The models' evaluations on the held-out test set resulted in 0.14 genome-wise MSE, 0.61 Pearson's R and 0.56 Spearman's R.

Although the Akita model has improved upon its predecessors by allowing for the predicting effects of DNA variants, the model's predictions are limited to 1-Mb windows of the genome and cannot be extended to model more distil long-range interactions and high-level structures like A/B-compartments.

## 1.2 Contribution

Our model improves upon the Akita model by replacing the dilated residual convolutions imputed in the "trunk" with the encoder of BERT, a bi-directional transformer. This change improves our ability to encode the long-range dependencies within the DNA sequence. We are then able to compute multiple predictions per DNA sample by using a variational layer with an evidence lower bound (ELBO) loss to model the data generatively. This results in our model producing the same benefits as the Akita model, but in addition, introducing a new range of benefits as it has the ability to compute intervals that capture uncertainty about some contact between $loci_i$ and $loci_j$.

## 2 Dataset

In our experiment, we used the same dataset on which Akita was pre-trained. The input data are human gene sequences with the length of $2^{20}$ base pairs (bp) ($\sim$ 1Mb), obtained from the chromatin by first fragmenting at assembly gaps, large unmappable regions or large filtered-out regions in contact maps, then extracting sequences within these fragments. The training set contains 7,008 sequences, extracted with the stride of $2^{18}$ bp ($\sim$ 262kb). The validation and test set contains 419 and 412 sequences respectively, both extracted with the stride of $2^{19}$ bp ($\sim$ 524kb). (Fudenberg et al. [2020])

For the target, five different types of chromain-chromain interaction maps are stacked for multi-task training. This includes three Hi-C datasets (GM12878, IMR90 and HCT116) (Rao et al. [2014]) and two Micro-C datasets (HFF Micro-C and H1hESC Micro-C) (Krietenstein et al. [2020]) A visualization of these contact maps for the same sequence is included in Figure 1. The datasets are preprocessed by the author of Akita by reprocessing into $2^{11}$bp (2048) bins, adaptively coarse-graining and normalizing the decrease in contact frequency due to increase in locus distance, applying some arithmetic operation, filling in missing bins with linear interpolation and applying Gaussian smoothing (`sigma=1`, `width=5`). This could effectively reduce the sparsity of the data and focus more on locus-specific interaction patterns. (Fudenberg et al. [2020])

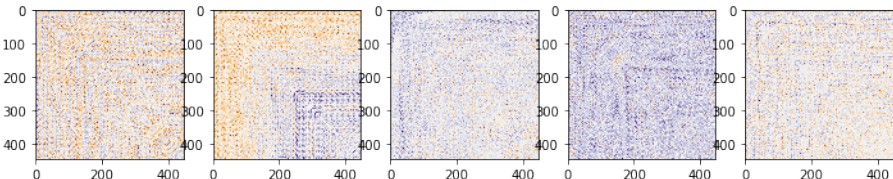

**Figure 1:** An example of DNA contact frequency maps. (From left to right: HFF Micro-C, H1hESC Micro-C, GM12878, IMR90 and HCT116 for the same sequence)

# 3 Methodology

Our methodology comprised of two parts: a reformulation of the Akita model using a bi-directional transformer and then a Bayesian extension on top of the transformer formulation. We did not change the "head" (decoder) of the Akita model[1]. Specifically, we replaced the dilated convolutions with the bi-directional transformer in the "trunk" (encoder), and then later added a variational layer after the bottleneck layer from the encoder. It's important to note that we focused on the encoder because we wanted to research the latent representations of the DNA sequences and their interactions. We first trained the transformer model (without the variational layer), tuning the hyperparameters separately, and then trained the model with the variational layer (the generative model) independently with another set of hyperparameters. Both sets of hyperparameters were determined manually using a binary search approach. We trained both models with SGD with momentum, both with Stochastic Weight Averaging (SWA) for model ensembling. The full hyperparamter configuration is available in the appendix under A.4. Finally, note that both models had a per layer time complexity of $\mathcal{O}(max(k \cdot n \cdot d \cdot f, \ n \cdot d^2))$, where $k, n, d, f$ are the convolution kernel size, sequence length, representation dimension, and number of convolution filters respectively. We based our running time analysis off of Vaswani et al. [2017].

## 3.1 A Transformer Formulation

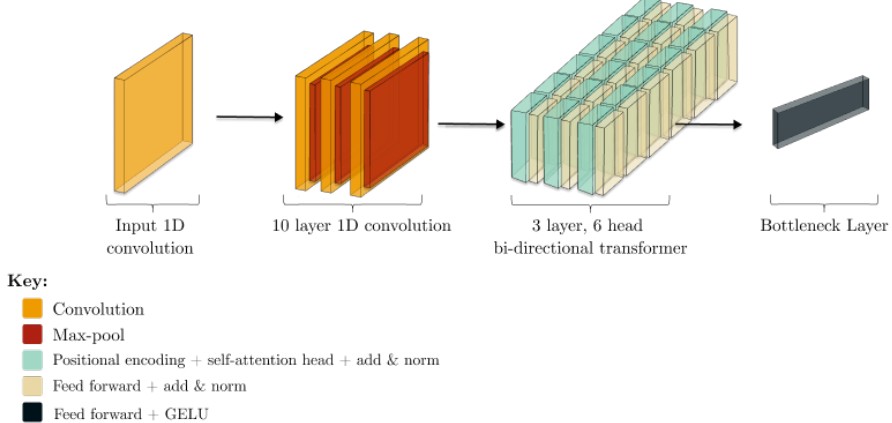

**Figure 2:** A not-to-scale diagram of the encoder with no variational layer.

As mentioned earlier, we replaced the dilated convolutions with a bi-directional transformer. The main motivating factor was the ability for self-attention to learn long range dependencies, which is analogous to what dilated convolutions aim to do by expanding the convolution's receptive field exponentially without losing resolution (Yu and Koltun [2016]). Improving on learning long range dependencies may be especially beneficial if a contact map contains many long range interactions, potentially leading to more accurate predicted maps. In building the bi-directional transformer, we stacked 3 layers of the so-called transformer encoder together, where each later had 6 attention heads (multi-head attention) and each feed forward layer had a dimension of 256. Note that the transformer configuration was shared between the generative and transformer only models.

Dissecting the transformer encoder layers, we can see that the self-attention modules are comprised of three key elements: the queries $Q$, keys $K$, and values $V$. Together, we computed the multi-head attention following Vaswani et al. [2017]:

$$\text{MultiHead}(Q, K, V) = \text{Concat}(\text{head}_1, ....\text{head}_h)W^O$$
$$\text{where head}_i = \text{Attention}(QW_i^Q, KW_i^K, VW_i^V)$$

Using this formulation we should shorten the length long range dependencies need to move in the network, potentially allowing for more learned granularity (Vaswani et al. [2017]). We did not change

---

[1]All full size model architecture diagrams are available in the appendix, under A.3

the other convolution layers in the encoder since we wanted to create compact embeddings of the DNA sequences for our transformer to learn better and felt that the convolutional layers from Akita were sufficient. Finally, we trained the model using MSE loss.

## 3.2 A Bayesian Extension

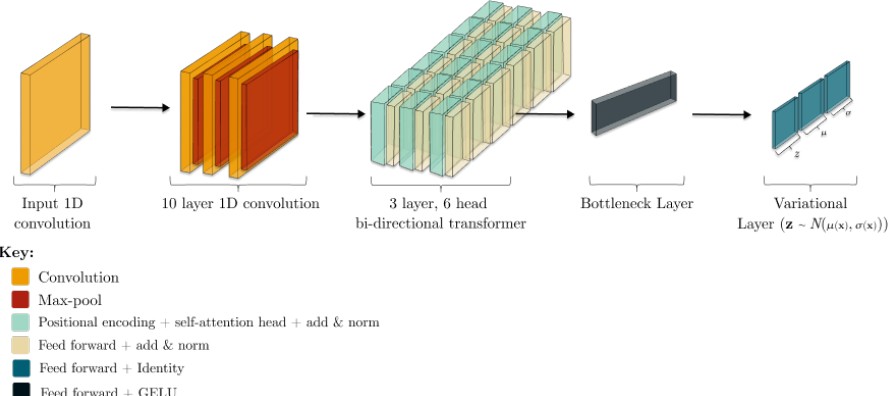

**Figure 3:** A not-to-scale diagram of the encoder with the variational layer.

The problem of predicting contact maps from DNA sequences potentially lends itself well to generative modelling, i.e., what if we could qualify the variability in predicted maps such that we could average the maps or examine all predicted maps for some sequence to better understand loci contact phenomena?

To accomplish this, we started from the usual definition of the so-called posterior predictive distribution, for some real-valued $N$-sample data $\mathcal{D} = \{(\mathbf{x}^{(1)}, \mathbf{y}^{(1)}), (\mathbf{x}^{(2)}, \mathbf{y}^{(2)}), ..., (\mathbf{x}^{(N)}, \mathbf{y}^{(N)})\}$:

$$p(\widetilde{\mathcal{D}}|\mathcal{D}) = \int_\theta p(\widetilde{\mathcal{D}}|\theta)p(\theta|\mathcal{D})\, d\theta$$

Note that $\theta$ above is normally the model parameters. But instead of averaging over the model parameters explicitly, we wanted to provide a simpler formulation owing to Akita's autoencoder nature and for potential computational reasons. Specifically, we instead formulate our goal as making our model additionally capable of averaging over the latent space, $\mathbf{z}$, to produce the predictive distribution. Mathematically that is,

$$p(\widetilde{\mathbf{y}}|\mathbf{x}, \mathcal{D}) = \int_{\mathbf{z}} p(\widetilde{\mathbf{y}}|\mathbf{x}, \mathbf{z})p(\mathbf{z}|\mathcal{D})\, d\mathbf{z}$$

Conveniently, we can consider $p(\widetilde{\mathcal{D}}|\mathbf{z})$ to be the decoder and $p(\mathbf{z}|\mathcal{D})$ to be the encoder. Also, this formulation allows us to create a predictive distribution without accounting for all model parameters. That helps computationally since it is easily possible for a network to be on the order of millions of parameters. Contrarily, the variational layer that contains the learned matrices for a sample of $\mathbf{z}$, $\mu_{\mathbf{z}}$ and $\sigma_{\mathbf{z}}$ are only $64 \times 64 = 4096$ parameters each (a total of 12228 parameters), entailing a great reduction in potential computational cost.

It follows then that we need to compute the posterior distribution $p(\mathbf{z}|\mathcal{D}) = \frac{p(\mathcal{D}|\mathbf{z})p(\mathbf{z})}{p(\mathcal{D})}$. Since computing the evidence ($p(\mathcal{D})$) is intractable, we will consider the KL-divergence between a variational distribution, $q(\mathbf{z}|\mathcal{D}) \sim \mathcal{N}(\mu_q(\mathbf{x}), \sigma_q(\mathbf{x}))$, that approximates the true posterior and the true posterior, $p(\mathbf{z}|\mathcal{D})$. More explicitly, we want to minimize the divergence formulated as $D_{KL}(q(\mathbf{z}|\mathcal{D}) \,||\, p(\mathbf{z}|\mathcal{D}))$. It eventually follows that we will use the ELBO of $\log(\mathcal{D})$ as our loss function, fit through maximum likelihood. The loss function can be better expressed as[2]:

---

[2]A full derivation is available in the appendix, under A.1

$$\log(\mathcal{D}) \geq \mathbb{E}_q[\log p(\mathbf{y}|\mathbf{x}, \mathbf{z})] - D_{KL}(q(\mathbf{z}|\mathcal{D}) \,||\, p(\mathbf{z}))$$

To further unpack the loss function, we see that the loss function is comprised of two terms, i.e., the expected data likelihood and the negative KL divergence term. For our model, we suppose that the likelihood is Gaussian, leading us to use MSE loss for that term. Ideally, we could approach expected likelihood through a Monte Carlo approach, i.e., taking many samples of $\mathbf{z}$ for some input DNA sequence, but due to practical computational limits, we took one sample from $\mathbf{z}$. Also while our decoder does not explicitly consider $\mathbf{x}$, we implicitly do as $\mathbf{z}$ is a function of the inputs ($\mathbf{x}$).

We also set the prior as $p(\mathbf{z}) \sim \mathcal{N}(0, \mathbf{I})$, and we did this for two reasons. First of which is that we followed Kingma and Welling [2014] as they used an isotropic Gaussian distribution in their paper introducing using variational Bayes in autoencoders, and it's fairly common to see this prior in literature. Second of which is that in the absence of a priori knowledge of the distribution of $\mathbf{z}$, we made the rather naive assumption that since $\mathbf{z}$ is a latent representation and because the targets are guaranteed to be on the interval (-2, 2) our prior would cover much of the density of the target distribution. This is of course not necessarily true, but it was our best guess absent of a priori knowledge. In practice, we saw the best performance using the isotropic Gaussian prior over tests of ~30 epochs with a zero mean and standard deviations ranging on the interval (0.965, 1.35)[3]. Since we assume $q(\mathbf{z}|\mathcal{D})$ to be a Gaussian distribution and because we set $p(\mathbf{z}) \sim \mathcal{N}(0, \mathbf{I})$, we have an analytic representation of our loss function used in training our final model[4]:

$$(\mathbf{y} - f(.))^2 + \sum_{j=1}^{J} 0.5(\log \sigma_{q,j}^2 - \sigma_{q,j}^2 - \mu_{q,j}^2 + 1)$$

Note that above $f(.)$ is the prediction from the decoder network, $J$ is the dimensionality of $\mathbf{z}$ and $\mu_{q,j}, \sigma_{q,j}$ comes from $q(\mathbf{z}|\mathcal{D}) \sim \mathcal{N}(\mu_q, \sigma_q^2)$.

Thus, we have formulated our model in a way such that we can extract multiple predictions for each latent encoding of a DNA sequence, allowing us to quantify the uncertainty in our predictions by way of examining the predictive distributions. Additionally we can approximate an expected prediction, i.e., $\mathbb{E}[p(\widetilde{\mathbf{y}}|\mathbf{x}, \mathcal{D})] = \frac{1}{N} \sum_{i=1}^{N} \widetilde{\mathbf{y}}_i$

## 4   Results

After approximately 40 meaningful (200 or more training steps) runs[5], we trained our two final models, and evaluated their abilities on various metrics. We evaluated our metrics on the sets of weights with the best validation MSE loss. We also compared the plots of predictions produced by the transformer versus target contact map and observed that the prediction could well approximate the pattern in the target. (Fig. 4)

|  | Transformer | VAE |
|---|---|---|
| MSE | 0.177 | 0.222 |
| Pearson's R | 0.396 | 0.081 |
| Spearman's R | 0.384 | 0.075 |

**Table 1:** Calculated metrics for our transformer and VAE models.

The difference in RMSE between our transformer model and Akita's model is $\sqrt{0.17} - \sqrt{0.14} = 0.038$, showing promise for our model.

---

[3] An analytic representation of the loss function for non-standard isotropic Gaussian priors is available in the appendix, under A.2

[4] A full derivation is available in the appendix, under A.2

[5] More detailed training notes in the appendix under A.5

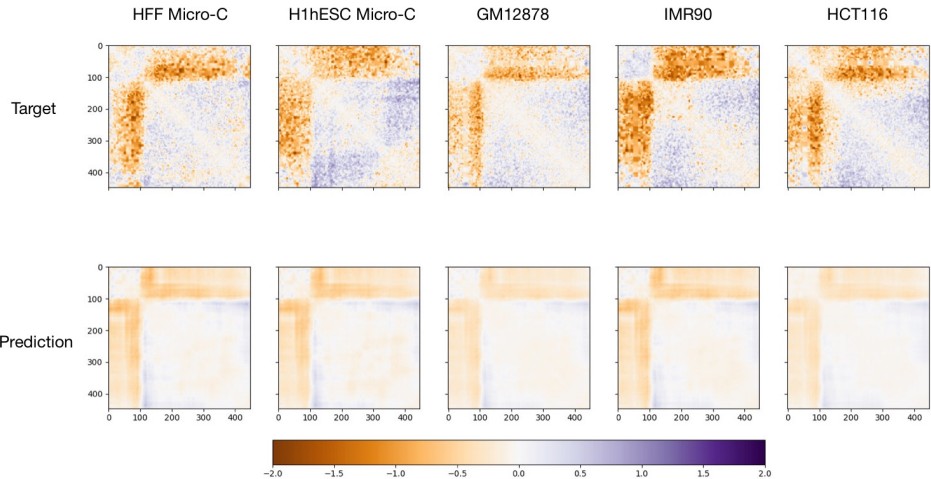

**Figure 4:** Target vs. prediction
6

Finally evaluating the generative model, we see that our framework and model opens up predicting 3D chromatin structure to fantastic new possibilities. We demonstrate the feasibility of averaging over the posterior predictive distributions through the comparison of the test set MSE loss between the generative model sampling just one prediction from the posterior predictive distribution, where it scored 0.2218 and sampling just 20 samples per sequence, achieving a MSE loss of 0.2217. Note that 20 samples is generally far under what would be recommended - this was a computational constraint. Additionally, samples from the posterior predictive distribution are able to well capture specific target values on the contact maps, this can be clearly seen by the plot in A.9 of the appendix. We also show that we do not have to average over the entire parameter space of the model to achieve decent results; keep in mind that we are only $\approx 0.09$, on absolute terms (RMSE), off of Akita's performance, which nears biological/technical replicates (Fudenberg et al. [2020]). As an added bonus, such a formulation allows us to have a really compact generative latent space - our model's latent space was only `batch_size` $\times 512 \times 4$ elements, far smaller than input space of `batch_size` $\times \approx 1$ million $\times 4$. Thus our approach and model is a feasible way of generative modelling for contact maps from DNA sequences, allowing for the quantification of uncertainty by generating distributions of predictions rather than point estimates and allowing for potentially better performance by averaging predictions over the posterior.

## 5 Discussion of Future Work

As model training and optimization was a big challenge in our research, future work could focus around better hyperparameter tuning strategies, e.g., Bayesian hyperparameter tuning. In particular, since the models showed significant overfitting around epoch 55, tuning the regularization and architecture could yield better results. Additionally, the choice of the prior was decidedly naive, so choosing a prior that better represent multi-modality of DNA interactions could an advisable next step, but it could be a step that requires significant biological research. Another potential improvement would be using more samples from $\mathbf{z}$ to feed into the decoder to better approximate the likelihood term, but that would be only advisable if powerful GPUs are available. Finally, another area of future research is constructing an architecture such that $\log p(\mathbf{x}|\mathbf{z})$ from (1.1) in the ELBO derivation can be included the in loss function.

---

[6]Another plot available in the appendix, under A.8

## 6  Conclusion

We proposed a model which utilizes transformer encoder to predict chromatin-chromatin interaction from DNA sequences. We conducted experiment on the dataset on which Akita was trained and our work has shown that attention mechanism works in terms of feature extraction from DNA sequences and is useful for capturing long-distance interaction within sequences. Additionally, by using our Bayesian formulation, we were able to create a model that demonstrates the feasibility of averaging over the latent space to predict contact maps - a departure from the usual VAE formulation that attempts to map the input space onto the input space. By having this be feasible, we demonstrate that we can now quantify the uncertainty in our prediction, certainly useful in tasks like engineered mutations, where there may not be ground truth.

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

# A   Appendix

## A.1   Derivation of ELBO

First suppose that we have a real-valued random latent vector $\mathbf{z}$ and a real-valued $N$-sample data $\mathcal{D} = \{(\mathbf{x}^{(1)}, \mathbf{y}^{(1)}), (\mathbf{x}^{(2)}, \mathbf{y}^{(2)}), ..., (\mathbf{x}^{(N)}, \mathbf{y}^{(N)})\}$.

We start the derivation of the ELBO loss function with the definition of the KL-divergence given our variational distribution $q(\mathbf{z}|\mathcal{D})$ and our true distribution $p(\mathbf{z}|\mathcal{D})$:

$$D_{KL}(q(\mathbf{z}|\mathcal{D}) \,||\, p(\mathbf{z}|\mathcal{D})) = \int_{\mathbf{z}} q(\mathbf{z}|\mathcal{D}) \cdot \log \frac{q(\mathbf{z}|\mathcal{D})}{p(\mathbf{z}|\mathcal{D})} d\mathbf{z} = \mathbb{E}_q[\log \frac{q(\mathbf{z}|\mathcal{D})}{p(\mathbf{z}|\mathcal{D})}]$$

Then using the expectation from above it follows:

$$\mathbb{E}_q[\log \frac{q(\mathbf{z}|\mathcal{D})}{p(\mathbf{z}|\mathcal{D})}] = \mathbb{E}_q[\log q(\mathbf{z}|\mathcal{D}) - \log p(\mathbf{z}|\mathcal{D})]$$

$$= \mathbb{E}_q[\log q(\mathbf{z}|\mathcal{D}) - \log \frac{p(\mathcal{D}, \mathbf{z})}{p(\mathcal{D})}]$$

$$= \mathbb{E}_q[\log q(\mathbf{z}|\mathcal{D}) - \log p(\mathcal{D}, \mathbf{z}) - \log p(\mathcal{D})]$$

$$= \mathbb{E}_q[\log q(\mathbf{z}|\mathcal{D}) - \log p(\mathcal{D}, \mathbf{z})] + \log p(\mathcal{D})$$

$$= -\mathbb{E}_q[\log p(\mathcal{D}, \mathbf{z}) - \log q(\mathbf{z}|\mathcal{D})] + \log p(\mathcal{D})$$

Then rearranging the above equation and substituting in the KL Divergence definition gets us:

$$\log p(\mathcal{D}) - D_{KL}(q(\mathbf{z}|\mathcal{D}) \,||\, p(\mathbf{z}|\mathcal{D})) = \mathbb{E}_q[\log p(\mathcal{D}, \mathbf{z}) - \log q(\mathbf{z}|\mathcal{D})]$$

$$\log p(\mathcal{D}) = \mathbb{E}_q[\log p(\mathcal{D}, \mathbf{z}) - \log q(\mathbf{z}|\mathcal{D})] + D_{KL}(q(\mathbf{z}|\mathcal{D})||p(\mathbf{z}|\mathcal{D}))$$

$$\geq \mathbb{E}_q[\log p(\mathcal{D}, \mathbf{z}) - \log q(\mathbf{z}|\mathcal{D})] \quad (1.0)$$

$$= \mathbb{E}_q[\log(p(\mathbf{y}, \mathbf{x}|\mathbf{z})) - \log q(\mathbf{z}|\mathcal{D})]$$

$$= \mathbb{E}_q[\log(p(\mathbf{y}|\mathbf{x}, \mathbf{z})p(\mathbf{x}|\mathbf{z})p(\mathbf{z})) - \log q(\mathbf{z}|\mathcal{D})]$$

$$= \log p(\mathbf{x}|\mathbf{z}) + \mathbb{E}_q[\log(p(\mathbf{y}|\mathbf{x}, \mathbf{z})p(\mathbf{z})) - \log q(\mathbf{z}|\mathcal{D})] \quad (1.1)$$

$$\geq \mathbb{E}_q[\log p(\mathbf{y}|\mathbf{x}, \mathbf{z})] + \mathbb{E}_q[\log p(\mathbf{z}) - \log q(\mathbf{z}|\mathcal{D})] \quad (1.2)$$

$$= \mathbb{E}_q[\log p(\mathbf{y}|\mathbf{x}, \mathbf{z})] + \int_z q(\mathbf{z}|\mathcal{D}) \log \frac{p(\mathbf{z})}{q(\mathbf{z}|\mathcal{D})} d\mathbf{z}$$

$$= \mathbb{E}_q[\log p(\mathbf{y}|\mathbf{x}, \mathbf{z})] - D_{KL}(q(\mathbf{z}|\mathcal{D}) \,||\, p(\mathbf{z})) \quad (1.3)$$

(1.0) The KL-Divergence is non-negative by concavity of the $\log(.)$ function and Jensen's inequality.
(1.1) Since we assume that $\mathbf{z}$ provides no information about $\mathbf{x}$ given the acyclic architecture of our model (and because $\mathbf{x}$ are the model inputs), we treat $p(\mathbf{x}|\mathbf{z})$ as an additive constant.
(1.2) Expression 1.1 must be greater than or equal to expression 1.2 since probability densities are non-negative.
(1.3) Substitute the definition of the negative KL-Divergence.

## A.2   Analytic representation of ELBO

First recall that we assume a Gaussian likelihood for the targets ($\mathbf{y}$), a Gaussian distribution for the variational distribution ($q(\mathbf{z}|\mathcal{D}) \sim \mathcal{N}(\mu_{\mathrm{q}}, \sigma_{\mathrm{q}}^2)$), and an isotropic Gaussian for the prior ($p(\mathbf{z}) \sim \mathcal{N}(\mu_{\mathrm{prior}}, \sigma_{\mathrm{prior}}^2)$). Finally note that we refer to predictions from our neural network as $f(.)$.

We start the derivation of the analytic representation of the ELBO from the result in A.1:

$$\mathbb{E}_q[\log p(\mathbf{y}|\mathbf{x},\mathbf{z})] - D_{KL}(q(\mathbf{z}|\mathcal{D}) \,||\, p(\mathbf{z})) = \mathbb{E}_q[\log p(\mathbf{y}|\mathbf{x},\mathbf{z})] - D_{KL}(q(\mathbf{z}|\mathcal{D}) \,||\, p(\mathbf{z}))$$

$$= \mathbb{E}_q[\log p(\mathbf{y}|\mathbf{x},\mathbf{z})] + \int_z q(\mathbf{z}|\mathcal{D}) \log \frac{p(\mathbf{z})}{q(\mathbf{z}|\mathcal{D})} d\mathbf{z}$$

$$= \mathbb{E}_q[\log p(\mathbf{y}|\mathbf{x},\mathbf{z})] + \mathbb{E}_q[\log p(\mathbf{z}) - \log q(\mathbf{z}|\mathcal{D})]$$

Since the KL-Divergence term in the above expression is complicated, we will examine it separately by first restating it as an expectation of a fraction:

$$\mathbb{E}_q[\log \frac{(\frac{1}{\sqrt{2\pi\sigma_{\text{prior}}^2}}\exp(-\frac{(z-\mu_{\text{prior}})^2}{2\sigma_{\text{prior}}^2}))}{(\frac{1}{\sqrt{2\pi\sigma_{\text{q}}^2}}\exp(-\frac{(z-\mu_{\text{q}})^2}{2\sigma_{\text{q}}^2}))}] = \mathbb{E}_q[\log(\frac{\sigma_{\text{q}}}{\sigma_{\text{prior}}}) - \frac{(z-\mu_{\text{prior}})^2}{2\sigma_{\text{prior}}^2} + \frac{(z-\mu_{\text{q}})^2}{2\sigma_{\text{q}}^2}]$$

$$= \log(\frac{\sigma_{\text{q}}}{\sigma_{\text{prior}}}) - \mathbb{E}_q[\frac{(z-\mu_{\text{prior}})^2}{2\sigma_{\text{prior}}^2}] + \mathbb{E}_q[\frac{(z-\mu_{\text{q}})^2}{2\sigma_{\text{q}}^2}]$$

$$= \log(\frac{\sigma_{\text{q}}}{\sigma_{\text{prior}}}) - \frac{1}{2\sigma_{\text{prior}}^2}\mathbb{E}_q[(z-\mu_{\text{prior}})^2)] + \frac{1}{2\sigma_{\text{q}}^2}\mathbb{E}_q[(z-\mu_{\text{q}})^2)]$$

$$= \log(\frac{\sigma_{\text{q}}}{\sigma_{\text{prior}}}) - \frac{1}{2\sigma_{\text{prior}}^2}\mathbb{E}_q[(z-\mu_{\text{prior}})^2)] + \frac{\sigma_{\text{q}}^2}{2\sigma_{\text{q}}^2}$$

$$= \log(\frac{\sigma_{\text{q}}}{\sigma_{\text{prior}}}) - \frac{1}{2\sigma_{\text{prior}}^2}\mathbb{E}_q[(z-\mu_q+\mu_q-\mu_{\text{prior}})^2] + \frac{1}{2}$$

$$= \log(\frac{\sigma_{\text{q}}}{\sigma_{\text{prior}}}) - \frac{1}{2\sigma_{\text{prior}}^2}(\sigma_q^2 + 0 + (\mu_q+\mu_{\text{prior}})^2) + \frac{1}{2}$$

$$= \log(\frac{\sigma_{\text{q}}}{\sigma_{\text{prior}}}) - \frac{\sigma_q^2 + (\mu_q+\mu_{\text{prior}})^2}{2\sigma_{\text{prior}}^2} + \frac{1}{2} \tag{1.0}$$

$$= \log \sigma_q - 0.5(\sigma_q^2 + (\mu_q+0)^2) + 0.5 \tag{1.1}$$

$$= 0.5(\log \sigma_q^2 - (\sigma_q^2 + \mu_q^2) + 1)$$

$$= 0.5(\log \sigma_q^2 - \sigma_q^2 - \mu_q^2 + 1) \tag{1.2}$$

(1.0) This expression is the one we used in our tests with non-unit isotropic Gaussian priors. Note that we sum over the dimensions of the vector $\mathbf{z}$ as done in Kingma and Welling [2014].
(1.1) We substitute in our choice of $\mu_{\text{prior}} = 0$ and $\sigma_{\text{prior}} = 1$.
(1.2) This is the expression we ultimately used in training our model. Note that again like (1.0) we sum over the dimensions of the vector $\mathbf{z}$ as done in Kingma and Welling [2014].

We can finally express the loss function for non-unit isotropic Gaussian priors with $J$ being the dimensionality of $\mathbf{z}$ as:

$$\mathbb{E}_q[\log p(\mathbf{y}|\mathbf{x},\mathbf{z})] + \sum_{j=1}^{J} \log(\frac{\sigma_{\text{q},j}}{\sigma_{\text{prior},j}}) - \frac{\sigma_{q,j}^2 + (\mu_{q,j}+\mu_{\text{prior},j})^2}{2\sigma_{\text{prior},j}^2} + \frac{1}{2}$$

$$\Longleftrightarrow$$

$$\frac{1}{N}\sum_{i=1}^{N}(\mathbf{y}_i - f(.)_i)^2 + \sum_{j=1}^{J} \log(\frac{\sigma_{\text{q},j}}{\sigma_{\text{prior},j}}) - \frac{\sigma_{q,j}^2 + (\mu_{q,j}+\mu_{\text{prior},j})^2}{2\sigma_{\text{prior},j}^2} + \frac{1}{2}$$

$$\Longleftrightarrow$$

$$(\mathbf{y} - f(.))^2 + \sum_{j=1}^{J} \log(\frac{\sigma_{\text{q},j}}{\sigma_{\text{prior},j}}) - \frac{\sigma_{q,j}^2 + (\mu_{q,j}+\mu_{\text{prior},j})^2}{2\sigma_{\text{prior},j}^2} + \frac{1}{2} \tag{1.3}$$

We similarly express the loss function we used to train our final model as:

$$\mathbb{E}_q[\log p(\mathbf{y}|\mathbf{x}, \mathbf{z})] + \sum_{j=1}^{J} 0.5(\log \sigma_{q,j}^2 - \sigma_{q,j}^2 - \mu_{q,j}^2 + 1)$$

$$\Longleftrightarrow$$

$$\frac{1}{N}\sum_{i=1}^{N}(\mathbf{y}_i - f(.)_i)^2 + \sum_{j=1}^{J} 0.5(\log \sigma_{q,j}^2 - \sigma_{q,j}^2 - \mu_{q,j}^2 + 1)$$

$$\Longleftrightarrow$$

$$(\mathbf{y} - f(.))^2 + \sum_{j=1}^{J} 0.5(\log \sigma_{q,j}^2 - \sigma_{q,j}^2 - \mu_{q,j}^2 + 1) \qquad (1.3)$$

(1.3) Recall that we only take one sample from the latent space per DNA sequence due to computational concerns, so $N = 1$ in the MSE calculations.

## A.3 Model Architecture Diagrams

The encoder (trunk) with no variational layer:

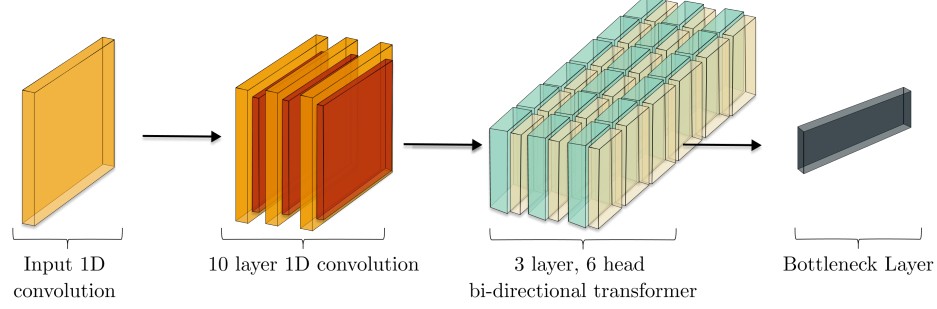

**Key:**
- ■ Convolution
- ■ Max-pool
- ■ Positional encoding + self-attention head + add & norm
- ■ Feed forward + add & norm
- ■ Feed forward + GELU

The encoder (trunk) with the variational layer:

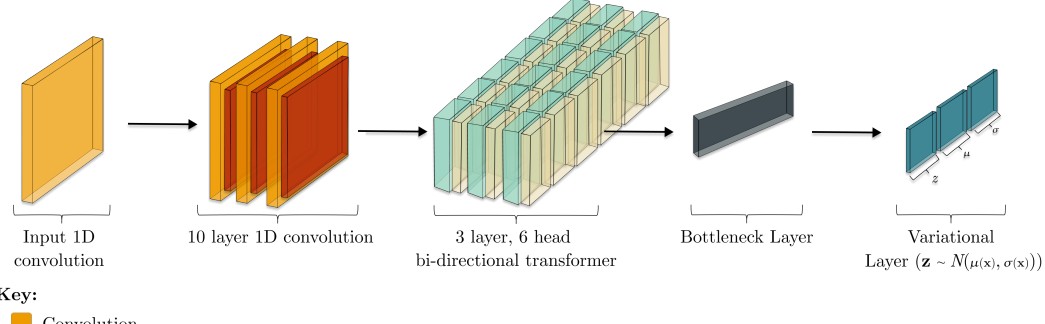

**Key:**
- ■ Convolution
- ■ Max-pool
- ■ Positional encoding + self-attention head + add & norm
- ■ Feed forward + add & norm
- ■ Feed forward + Identity
- ■ Feed forward + GELU

The decoder (head):

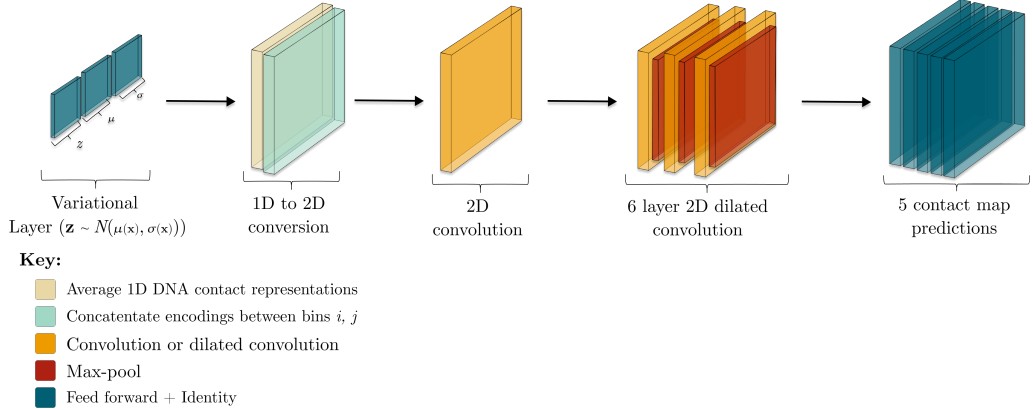

**Key:**
- ▢ Average 1D DNA contact representations
- ▢ Concatentate encodings between bins $i, j$
- ▢ Convolution or dilated convolution
- ▢ Max-pool
- ▢ Feed forward + Identity

Note that the variational layer is included here to show the inputs into the decoder. There is only one variational layer in the entire model (encoder and decoder).

## A.4   Hyperparameters

**Transformer only model**:

`SGD learning rate`: 0.0045
`SGD momentum`: 0.975
`Patience`: 40
`Stochastic weight averaging annealing epochs`: 5
`Stochastic weight averaging learning rate`: 0.00085
`Max epochs`: 65
`Transformer dropout`: 0.15
`Stochastic shift amount`: 11
`Gradient clip value`: 10.7
`Batch size`: 2

**Generative model**:

`SGD learning rate`: 0.004
`SGD momentum`: 0.975
`Patience`: 40
`Stochastic weight averaging annealing epochs`: 5
`Stochastic weight averaging learning rate`: 0.0025
`Max epochs`: 65
`Transformer dropout`: 0.1
`Stochastic shift amount`: 11
`Gradient clip value`: 10.7
`Batch size`: 2

## A.5   Training Notes

We first started training the model using an RTX 3080, but ultimately there were significant issues in doing so as we could only fit one batch on it, resulting in poor model performance. There was approximately 35 training runs using the RTX 3080 but these runs ultimately weren't really insightful for the final model. Eventually, we switched to 4 Titan X or Titan Xp GPUs. These allowed us to train with 2 batches due to an increased amount of vram. These however were rather slow; the transformer model took 5 days to train for the final model while the variational model early stopped after 48 epochs, taking approximately 2 and a half days. After switching to these GPUs we found that the Adam optimizer performed much worse than the SGD with momentum optimizer.

## A.6 Definitions of Metrics

Like Akita, we used MSE, Pearson's R, and Spearman's R to quantify the performance of our models.

$$\text{MSE} = \frac{1}{n} \sum_{i=1}^{n} (y_i - \hat{y}_i)^2$$

where $n$ is the number of data points, $y_i$ are the predicted values, and $\hat{y}_i$ are the target values in the test set.

$$\text{Perason's R} = \frac{\sum (x_i - \overline{x})(y_i - \overline{y})}{\sqrt{\sum (x_i - \overline{x})^2 \sum (y_i - \overline{y})^2}}$$

where $x_i$ are the values in the upper triangular portion of the predicted contact maps, $\overline{x}$ is the mean of all the $x_i$, $y_i$ are the values in the upper triangular portion of the target contact maps, $\overline{y}$ is the mean of all the $y_i$.

$$\text{Spearman's R} = 1 - \frac{6 \sum d_i^2}{n(n^2 - 1)}$$

where $n$ is the number of data points, which is the number of values in the upper triangular portion of the contact maps, and $d_i$ is the difference between the ranks of two observations.

## A.7 Table of MSE, Pearson's R, and Spearman's R for Each Dataset

|  | HFF | H1ESC | CM12878 | IMR90 | HCT116 |
|---|---|---|---|---|---|
| MSE | 0.22 | 0.18 | 0.13 | 0.23 | 0.11 |
| Pearson's R | 0.42 | 0.44 | 0.37 | 0.38 | 0.38 |
| Spearman's R | 0.40 | 0.43 | 0.35 | 0.37 | 0.37 |

**Table 2:** Calculated metrics for each of the five datasets for our transformer model.

|  | HFF | H1ESC | CM12878 | IMR90 | HCT116 |
|---|---|---|---|---|---|
| MSE | 0.29 | 0.23 | 0.16 | 0.29 | 0.14 |
| Pearson's R | 0.09 | 0.08 | 0.09 | 0.08 | 0.06 |
| Spearman's R | 0.09 | 0.08 | 0.08 | 0.07 | 0.05 |

**Table 3:** Calculated metrics for each of the five datasets for our VAE model.

## A.8    Transformer prediction

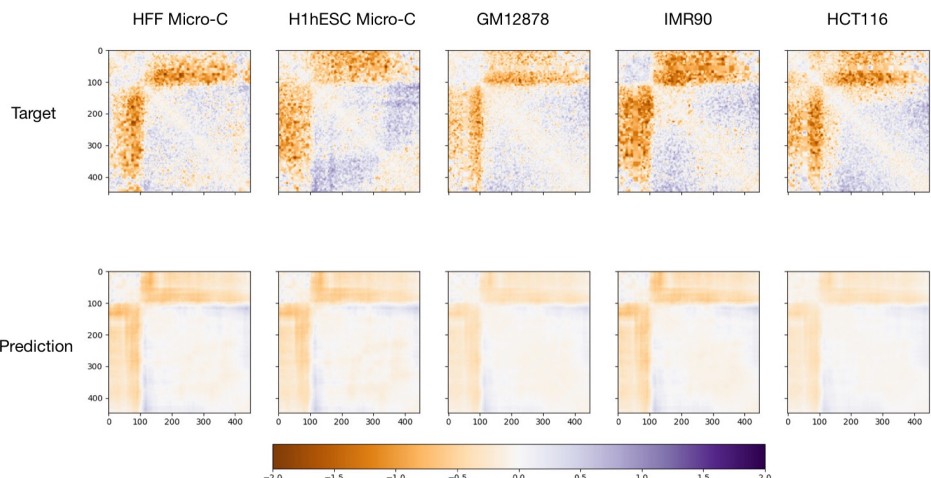

## A.9 VAE Prediction Distribution Visualization vs Target Value

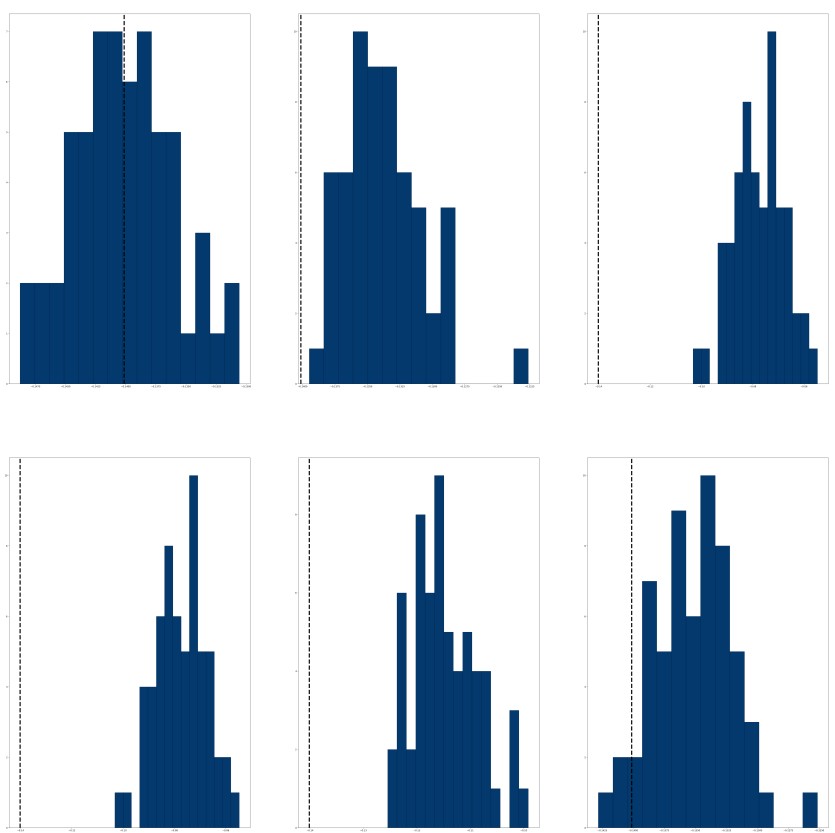

