# OpenReview forum: "Attention to Bayes: Self-Attention and Bayesian Methods for Predicting 3D Chromatin Folding From DNA Sequences"
_uoft.ai/University_of_Toronto/2021/ProjectX — Submitted to ProjectX2021_

### Official Review · Reviewer_a7wz · 2022-02-10
**A review of Attention to Bayes: Self-Attention and Bayesian Methods for Predicting 3D Chromatin Folding From DNA Sequences**

**Rating:** 5
**Confidence:** 4

**Review:**

The authors present an extension to a work in a well established biomedical field: predicting epigenetic markers directly from sequence.  In this case, the task is to predict 3D chromatin-mediated contact within 1mb segments of the genome.

The authors do a good job covering relevant literature, especially Fudenberg et al. [2020], which the paper is heavily based on and compared to.  There are two core ideas:

1. replace dilated convolutions with a bidirectional transformer architecture, in the hopes of better capture of long-range interactions
2. add a variational layer, to enable the prediction/calculation of uncertainty in the predictions.

While the authors were successful in doing both of these, it's not clear that it made any difference.

First, the performance of the model was worse or potentially equivalent to Fudenberg et al. showing that at least for the range of hyperparameters tested, there was not a jump in performance.  This is not terribly surprising as dilated convolutions have been shown to be quite effective in this context, and the authors did not alter the datasets to increase the extent of the input past 1mb, which might have highlighed a difference.  Furthermore, there was no analysis of the attention patterns learned by the transformer.  Maybe this was due to time constraints, and could be explored in extensions to this work.

Secondly, the variational layer seemed to only decrease the fidelity of the prections.  In the single example shown, there are many subtleties of the targets that are not captured (even in a "blury" way), and indeed, any differences between the different target tasks were minimal.  There was no analysis of the uncertainty of the samples, just plots of an average, nor was there any analysis of the latent representations.  Both of these should be considered as part of further work in this area.

Overall, while the idea is appealing from first principles, the paper is little more than a proof-of-concept that these alterations to the Akita network more or less work, but there is not enough analysis or justification of these changes thus far to say anything of significance.

---

### Official Review · Reviewer_QUj3 · 2022-02-12
**The authors propose to use Transformers and VAE with self-attention modules to predict contact maps from DNA sequence**

**Rating:** 7
**Confidence:** 4

**Review:**

**Connection to Current Science (science and practice)**: 2

**Clarity of Communication**: 1.5

* schematic would have been useful in introduction. For example, a visual that shows what type of long range interactions the attention is intended for.
* Also the “Input 1D convolution” (orange) in A.3 is confusing. What is 1D about it? Why is it a 2d array?

**Methodological Quality**:3

* idea of using attention seems very reasonable
* There is a literature on VAEs that you may not be aware of. e.g. IWAE https://arxiv.org/abs/1509.00519

**Reproducibility**: 0

* not only is there no code, but it is not clear what software you implemented things in. It’s a major oversight since your A.4 Hyperparameters are not that useful or informative without mentioning these details.

---

### Official Review · Reviewer_jKY4 · 2022-02-12
**The goal of this study is to develop a new deep learning-based approach that combines transformers and variational autoencoder (VAE, I guess since the paper has never defined what VAE is) to take long-range of interactions or dependencies within DNA sequence for predicting 3D chromatin folding. Overall, the motivation and justification of the model are suitable. There are also some novel ideas in formulizing the algorithm.**

**Rating:** 8
**Confidence:** 5

**Review:**


Major strengths
-Use the attention mechanism in deep learning to model the long-range dependence in DNA sequence.
-Use a generative model to predict the 3D chromatin folding with confidence measurement

Major limitations
-Although the model formulation looks to be elegant, its performance is not that great. For example, the correlation coefficients shown in Table 1 are still low, even much lower than those reported by the Akita model.  The rationale for this should be discussed.
-It has quite good potential to add an interpretable component in the model architecture based on its outputs, such as the uncertainty measurements. This will be an interesting direction for this model.
-The new model should be compared with more recent state-of-the-art models for 3D chromatin folding prediction since this is a quite active area in recent years and there are many elegant methods developed in recent years.
-The description of the variable Y measuring the 3D chromatin folding can be clearer.  This will help readers of the manuscript understand the research question, especially for those who have no background in biology.
-Since one of the main advantages of the model is to measure the long-range dependence, the performance of the different sizes/lengths of the ranges or windows should be tested.
-Figure 1 is missing legends of the colors.

---

### Official Review · Reviewer_FSgh · 2022-02-15
**Promising but ambiguous results.**

**Rating:** 4
**Confidence:** 3

**Review:**

This paper presents a new architecture for predicting genome folding from DNA.  Specifically, the model adapts the encoder of "Akita" model by incorporating self-attention mechanisms in the form of three layers of multi-head, bidirectional transformer in lieu of the original fully convolutional architecture.  The modification was motivated by the need to capture and represent long-range interactions and dependencies in the sequence.  The reported results suggest that the model performs slightly worse than the original CNN-based model (e.g., MSE of 0.177 vs 0.14) however it's unclear if this is due to a fundamental problem with the architecture or with computational limits of the experimental set (e.g., due to limited training time, lack of model averaging, etc).

The paper also proposes incorporating a VAE in hopes of producing multiple output samples.  Empirically this performed worse than the straight prediction approach for the primary task and the ability to draw and explore multiple outputs wasn't explored in the results.

The paper is reasonably written but unclear in parts.  For instance, certain notation is never defined (e.g., $\tilde{D}$, $\tilde{y}$).  It's only implied that $x$ is the input DNA sequence and $y$ is the output interaction maps.

Overall the idea of leveraging attention for this problem is interesting and appropriate.  However, based on the results of this paper, it remains unclear of it's value.  The results are promising but critical comparisons are missing, e.g., a more comparison to a CNN-only architecture.  The original Akita results are better than those reported with attention but the full Akita model has other benefits (e.g., model averaging) which weren't able to be explored in this paper.  To be concrete, it would be good to see, using the authors experimental and training setup, the results of the original CNN architecture reported in Table 1.

---

### Decision · Program_Chairs · 2022-02-19

NA